# Nanoparticle-Based Interventions for Liver Transplantation

**DOI:** 10.3390/ijms24087496

**Published:** 2023-04-19

**Authors:** Joseph Sushil Rao, Robert Ivkov, Anirudh Sharma

**Affiliations:** 1Division of Solid Organ Transplantation, Department of Surgery, University of Minnesota, Minneapolis, MN 55455, USA; 2Schulze Diabetes Institute, Department of Surgery, University of Minnesota, Minneapolis, MN 55455, USA; 3Department of Radiation Oncology and Molecular Radiation Sciences, Johns Hopkins University School of Medicine, Baltimore, MD 21231, USA; 4Department of Oncology, Sydney Kimmel Comprehensive Cancer Center, Johns Hopkins University School of Medicine, Baltimore, MD 21287, USA; 5Department of Mechanical Engineering, Whiting School of Engineering, Johns Hopkins University, Baltimore, MD 21218, USA; 6Department of Materials Science and Engineering, Whiting School of Engineering, Johns Hopkins University, Baltimore, MD 21218, USA

**Keywords:** nanoparticles, hyperthermia, liver transplant, tolerance

## Abstract

Liver transplantation is the only treatment for hepatic insufficiency as a result of acute and chronic liver injuries/pathologies that fail to recover. Unfortunately, there remains an enormous and growing gap between organ supply and demand. Although recipients on the liver transplantation waitlist have significantly higher mortality, livers are often not allocated because they are (i) classified as extended criteria or marginal livers and (ii) subjected to longer cold preservation time (>6 h) with a direct correlation of poor outcomes with longer cold ischemia. Downregulating the recipient’s innate immune response to successfully tolerate a graft having longer cold ischemia times or ischemia-reperfusion injury through induction of immune tolerance in the graft and the host would significantly improve organ utilization and post-transplant outcomes. Broadly, technologies proposed for development aim to extend the life of the transplanted liver through post-transplant or recipient conditioning. In this review, we focus on the potential benefits of nanotechnology to provide unique pre-transplant grafting and recipient conditioning of extended criteria donor livers using immune tolerance induction and hyperthermic pre-conditioning.

## 1. Introduction

Organ and tissue recovery for transplantation involves an inevitable ischemic event when blood is replaced with a cold preservation solution [1]. Cold preservation (0–4 °C) terminates oxygenation to the organ, generating metabolic waste and depleting energy stores, ultimately leading to an increasingly acidotic environment [2,3]. Following cold preservation, ischemic injury is exacerbated by reperfusion when the blood supply is restored at the time of transplantation. Ischemia-reperfusion injury (IRI) is characterized by reactive oxidative intermediates, which manifest and amplify ischemic damage through a cascade of metabolic stresses and innate immunological responses, causing cellular damage [1,4].

Liver transplantation is the standard of care for end-stage liver disease [5]. The global organ shortage obligates centers to use extended-criteria donor livers [6] and organs subjected to longer cold ischemia exposure and/or longer warm ischemia exposure through donation after circulatory death. Extended criteria donor livers, which include livers from older donors, steatotic livers and those donated after cardiac death with longer warm ischemia times (DCD) [6], are susceptible to greater IRI, contributing to more limited graft function [7,8]. Livers exposed to longer cold ischemia may present with endothelial cell swelling [9], platelet aggregation in sinusoids, vasoconstriction and microcirculatory failure during the early postoperative period [9,10,11], often resulting in primary graft non-function if the underlying IRI is not addressed. Technologies aiming to minimize IRI could improve graft function by shortening recovery time, enabling the use of extended criteria donor livers.

IRI can be classified into warm and cold types [8,12]. Warm IRI is initiated by hepatocellular injury and develops in situ, soon after reperfusion, with the potential for acute hemodynamic responses causing severe morbidity and even mortality. Cold IRI occurs before transplantation during cold preservation, affecting the sinusoidal endothelial cells, disrupting the microenvironment, and often exacerbating warm IRI at the time of transplantation. Although the insults differ, both types of IRI begin with local innate immune inflammatory responses.

When activated, Kupffer cells and neutrophils release cytokines and chemokines that stimulate the production of reactive oxygen species (ROS). This cascade increases oxidative stress, stimulates the expression of adhesion molecules and supports the infiltration of lymphocytes and monocytes [7,13]. Unlike an alloimmune response, the innate response to IRI in livers is mediated by the pattern recognition receptor (PRR) system. Injury associated with hepatocytes initiates a damage-associated molecular pattern (DAMP) recognition response to tissue inflammation. The adaptive immune response in the liver depends primarily on CD4^+^ T cells [14]. In addition, innate to adaptive immune signaling can contribute to an acute inflammatory state [8] while the liver recovers from cold preservation.

Following IRI, an alloimmune response arises from an adaptive immune response [15,16,17] through the conventional T cell-mediated response to alloantigens presented by the donor organ that determines graft survival. Although the acute rejection of liver grafts is uncommon, the induction of immunosuppression in liver transplantation improves graft survival assessed at 3 and 12 months and 5 years [18]. Quadruple and triple regimens consisting of low-dose calcineurin inhibitors (cyclosporine and tacrolimus), mycophenolate mofetil (MMF), IL-2RA and steroids with and without mTOR inhibitors (mTORi) effectively induce immunosuppression [19]. However, trough levels of tacrolimus are important to monitor to prevent nephrotoxicity. Switching to monotherapy with either MMF or mTORi has been associated with reduced incidence of graft loss to rejection [18,20]. Although immunosuppressants have demonstrated lower graft loss, inadvertent side effects such as nephrotoxicity (tacrolimus), wound healing impairment (tacrolimus, sirolimus, everolimus), lymphomas (most immunosuppressants), skin malignancies (cyclosporine) and thrombocytopenia (tacrolimus) [19] could be avoided with novel tolerance induction protocols.

## 2. Organ Preconditioning to Improve Allocation of Marginal Livers for Transplant

Organ preconditioning offers tremendous potential to increase the pool of transplantable livers. Current preconditioning methods include pharmacological reconditioning and gene therapy [21,22,23,24] using ex vivo normothermic perfusion after donation. The general principle is to expose the hepatocytes and non-parenchymal cells to sublethal stress and provide them the opportunity to recuperate from hypothermia or other cellular injuries before they are transplanted [25]. These preconditioning protocols have demonstrated benefits such as reduced post-transplant transaminases, necrosis and apoptosis of sinusoidal endothelial cells, lower leukocyte activation and increased tolerance to hypoxia by hepatocytes, with an overall improved graft survival [25].

Protective effects conferred by these techniques correlate with the upregulation of heat shock proteins (HSPs), e.g., HSP27, HSP72 and HSP90 [26,27,28]. HSPs are a class of evolutionarily conserved proteins across species and cell types that are induced by a wide range of stresses and stimuli [29,30]. HSPs generally act as molecular chaperones, in part to confer protective effects to the cell from thermal or mechanical injury by facilitating protein (re)folding and ensuring the fidelity of downstream cellular signaling [29,30]. For example, the upregulation of HSP70 in many cancers can facilitate resistance to different treatment modalities, including hyperthermia, by imparting thermotolerance. Consequently, HSP70 has become a target for the development of some therapies [31]. On the other hand, for liver transplantation, protection imparted by HSP upregulation is desirable to prevent damage from IRI. Current preconditioning techniques involving chemotherapeutics targeting HSP induction suffer from a lack of specificity and induce toxic side effects. Thus, new approaches are needed that offer targeted preconditioning of the organ or transplant site. In the subsequent sections, we review approaches that exploit nanotechnology to initiate liver-specific immune modulating responses and mild hyperthermia treatment with magnetic nanoparticle hyperthermia of the graft (Figure 1), intended to initiate tolerance mechanisms, including controlled HSP70 upregulation, that may prove to be viable alternatives.

### 2.1. Expanding the Donor Pool by Inhibiting IRI with Nanotechnology

Liver-specific immunomodulatory protocols using nanoparticles can be used to expand the donor pool by conditioning donor livers, minimizing damage from IRI and improving tolerance induction. Technologies, including receptor-specific nanoparticle targeting-based immunomodulation and recipient tolerance induction using nanoparticles, will be discussed. These efforts towards nanoparticle-based pre-liver transplantation therapies could potentially improve short- and long-term outcomes.

The ‘injury hypothesis’ states that IRI activates a cascade of proinflammatory responses dominated by innate immune cell phenotypes (myeloid-derived) that trigger an adaptive immune cascade culminating in allograft rejection [32]. The PRRs are primarily expressed on the surface of macrophages and dendritic cells and lead to the transcription of genes associated with immune responses when they are activated [33]. Among the PRRs, Toll-Like Receptors (TLRs) are predominantly involved in the liver IRI cascade. The TLR system consists of homodimers or heterodimers of type I transmembrane glycoproteins present on the cell membrane and membrane of the endoplasmic reticulum. TLR ligation by binding of lipopolysaccharides triggers intracellular signaling pathways resulting in the activation of transcriptional factors initiating the expression of genes encoding cytokines, chemokines and co-stimulatory molecules [34,35,36].

TLR-4 is specific to liver IRI. Mice deficient in TLR-4 were protected from IRI with suppressed local hepatic inflammation [37,38]. In addition, blocking TLR-4 signaling on both liver non-parenchymal cells and parenchymal cells is critical in reducing IRI [39]. Nanoparticles perfused into the liver can be used to block TLR-4 receptor activity by binding without initiating their signaling cascade with the help of HMGB1, a key endogenous ligand required for the activation of IRI in the liver [40].

HMGB1 is released from injured hepatocytes to stimulate liver non-parenchymal cells such as Kupffer cells via TLR-4 signaling [41]. In addition, hypoxic hepatocytes release HMGB1 facilitated by TLR4-dependent production of ROS, and in turn, ROS induces HMGB1 through CaMK-dependent mechanisms encouraging a sustained inflammatory response during liver IRI [13]. The successful blockade of this cascade with nanoparticles can prevent the initiation of a pro-inflammatory cascade, thus preventing IRI.

Involvement of receptors for advanced glycation end products (RAGE) has been well described in IRI by regulating the expression of CXCL2 (macrophage inflammatory protein 2) via epidermal growth factor receptor (EGFR)-dependent mechanisms [42], which can also be blocked by nanoparticles. Magnetic nanoparticles can provide targeting validation using high-resolution imaging techniques such as magnetic particle imaging (MPI), co-registered with magnetic resonance imaging (MRI), X-ray computed tomography (CT) and other anatomical imaging modalities.

Although a specific receptor blockade with nanoparticles can limit an IRI cascade, it is unlikely that all pathways of IRI activation can be blocked successfully by nanoparticle-based receptor binding technology. T cell co-stimulatory pathways promote IRI through alloantigens. These pathways can be blocked by cytotoxic T-lymphocyte antigen 4 immunoglobulin (CTLA4Ig) [43]. Nanoparticles targeted at CTLA4Ig could further enhance nanoparticle-based IRI blockades, effectively preventing non-allogenic T cell responses to IRI. The co-stimulatory T cell function of CD154 on liver T cells is also critical to activate the innate immune system [8] and is a potential target for nanoparticle-based receptor blockade.

The T cell immunoglobulin mucin-containing molecule-1 (TIM-1) is expressed primarily on CD4 T_H_1 and T_H_2 cells, and TIM-4 is expressed mostly on macrophages and other antigen-presenting cells. TIM-1 and TIM-4 interactions have demonstrated a T cell–macrophage regulation at the innate-adaptive immune interface, which could be a target site for nanoparticle binding [44]. Nanoparticles targeting TIM-1 may ameliorate hepatocellular damage in ischemic livers.

Liver-specific immunomodulatory protocols using nanoparticles can be used to expand the donor pool by conditioning donor livers, minimizing damage from IRI and improving tolerance induction. These efforts towards nanoparticle-based pre-transplantation therapies could potentially improve short- and long-term outcomes.

### 2.2. Nanoparticle-Based Tolerance Induction through Donor Graft Preconditioning

Another suggested nanoparticle-based preconditioning method targets endothelial cells in the vasculature of the donor graft, which express non-self MHC molecules and display an increased antigen-presenting capacity (upregulated (ICAM)-1 and (VCAM)-1 on IRI. They represent an interface between the recipient and donor following transplantation and, thus, a site for strong recipient immune activation and, ultimately, allograft rejection. Here, allorecognition by recipient antigen-presenting cells followed by recruitment of cytotoxic and memory T cells can initiate graft rejection. However, an intact vascular endothelium is essential for maintaining physiological pressures, flow resistance and long-term graft stability post-transplantation. Thus, non-destructive strategies targeting graft endothelium are employed, including gene silencing, blocking cell surface receptors that mediate leukocyte recruitment and release of tolerogenic drug molecules intracellularly following receptor-mediated endocytosis by endothelial cells [45,46,47,48] (Table 1).

Nanoparticle drug carriers can provide receptor-mediated cell targeting through the coating of ligands on the nanoparticle specific to cell receptors, endocytosis after specific targeting and selective release of drugs to the endothelium [48,49,50] (Table 1). The use of nanoparticles also provides image-guided (MRI and MPI) targeting validation [51,52,53,54,55]. Thus, targeting endothelial cells through ex vivo perfusion of coated nanoparticle drug carriers is an attractive strategy that can reduce reperfusion injury, initiating an innate immune response to improve graft survival of marginal donors.

For e.g., Cui et al. showed that Poly (amine co-ester) nanoparticles loaded with non-self MHC II specific siRNA and delivered in a rat through ex vivo perfusion resulted in the attenuation of MHC II molecules, reduced T cell infiltration and T cell-mediated inflammation and improved allograft histology [50]. Zhu et al. showed that targeted endocytosis of Rapamycin (tolerogenic drug) loaded polyethylene glycol micelles by endothelium (via αVβ3 integrins) in mouse aortic and tracheal allografts reduced the secretion of inflammatory cytokines and prevented allograft rejection post-transplantation [56]. Additionally, a dose enhancement with a 10-fold lower rapamycin dose was required to inhibit allograft rejection when Rapamycin was loaded in nanoparticle micelles vs. free Rapamycin [56] (Table 1). The advantage of ex vivo perfusion-based cell-receptor targeting is that true targeting can be achieved in a short time vs. systemic administration, which is confounded by a host of immunological and physiological variables. Although ex vivo methods have demonstrated success in rodent models, successful scale-up of the methods needs to be validated in large animal models (porcine, rabbits, etc.) before translating to humans. Magnetic nanoparticles can aid in such scaled-up validation and in mechanistic studies because they enable various imaging modalities (MRI, dual energy-CT and MPI) [57,58].

### 2.3. Tolerance Induction through Recipient Conditioning with Nanoparticles

Generally, strategies in allogenic tolerance induction are focused on recipient immunoregulation by targeted deletion or expansion of recipient immune cell sub-populations (cytotoxic T cells (CTLs), regulatory T cells (T_reg_), B cells and macrophages). T_reg_ subpopulations of T cells are necessary for the suppression of self-reactive T cells in the periphery. Often, a balance is sought between the deletion of peripheral (non-thymic) alloantigen-specific CD8^+^ T cells and the expansion of CD4^+^ CD25^+^ T_regs_. This is termed peripheral tolerance induction [59,60]. Although other T cell subsets play a role in tolerance induction, the role of T_reg_ is recognized as dominant and is well characterized in preclinical and clinical models [60,61,62,63,64]. In recipients with donor-specific tolerance, T_reg_-based immunoregulatory networks are highly active [63]. Additionally, various tolerance induction treatments fail when T_reg_ are absent [63,64]. The prospects of immediate graft tolerance by targeted depletion of alloreactive T cells and the pre-generation of allogenic T_reg_ in the recipient prior to transplantation are appealing and could circumvent the need for long-term immunosuppression.

A second strategy to induce allogenic tolerance is central tolerance. Central tolerance is a non-response tolerance state of the body to autoantigens during the immature embryonic stage of development of the immune system and the development of T and B cells in the central immune organs. Genes that encode the T cell receptor (TCR)/B cell receptor (BCR) rearrange to produce variants of TCR/BCR that recognize different antigens. The induction of chimerism or central tolerance happens in the thymus of the recipient by ablation of the recipient’s immune system, followed by the administration of donor bone marrow cells. APCs from donor bone marrow populate the recipient thymus and selectively delete developing alloantigen-reactive recipient CTLs, a process called negative selection [59,60]. This mechanism is especially interesting in the context of liver transplantation because it has been shown, at least preclinically, that liver allografts are successfully transplanted in recipients without the need for long-term immunosuppressants [65,66]. While underlying mechanisms are still being investigated, this observation has sometimes been attributed to liver-specific “microchimerism,” where donor liver APCs and soluble MHC class I molecules selectively delete alloreactive CD8^+^ T cells in the recipient following transplantation and reperfusion [64,67]. Additionally, the induction of T_reg_ has been concurrently observed in this process [68,69], which might be induced by liver sinusoidal endothelial cells (LSECs) and liver-specific dendritic cells.

The liver microenvironment is highly conducive to the induction of immune tolerance and is worth investigating further. Nanoparticles have demonstrated potential for both strategies in preclinical models to improve targeting efficiency with dose enhancement, i.e., much lower doses of nanoparticle-coated donor antigens and peptides are needed compared to free peptides to tolerize recipient immune cells or induce chimerism, thus minimizing (sometimes fatal) side effects of systemic exposure high concentrations of peptides [70,71]. In the following discussion, we review the application of nanoparticle carriers to induce donor antigen-specific tolerance in recipients by targeting recipient myeloid cells and through direct recipient alloantigen-specific CD8^+^ T cell targeted depletion and allogenic T_reg_ expansion.

Recently, nanoparticles have been exploited to induce antigen-specific tolerance as they are able to carry multiple functional molecules simultaneously through surface-coating, including immune cell-targeting ligands, donor antigens and immunomodulatory drugs. Two strategies discussed previously have been adopted for tolerance induction in recipients—(i) peripheral tolerance induction and (ii) central tolerance induction through mixed chimerism. In peripheral tolerance induction, strategies targeting recipient antigen-presenting cells and alloreactive CD4^+^ and CD8^+^ T cells have been tested. The ability of donor antigen-labeled nanoparticles to target cell receptors of recipient antigen-presenting cells (APCs) and selectively deliver tolerogenic therapeutics (e.g., Rapamycin) to “tolerize” the recipient APCs to donor allograft antigens makes them an attractive technology to induce donor-antigen specific peripheral tolerance.

Stead et al. showed that dendritic cells (DCs) harvested from OVA-sensitized C57/Bl/6 mice could be targeted in the spleen and peripheral blood by the i.v. injection of porous silicon nanoparticles coated with DC-specific intercellular adhesion molecule-3 grabbing non-integrin (DC-SIGN), monoclonal antibody CD11c, ovalbumin (OVA) and tolerogenic payload, Rapamycin, to upregulate donor-specific regulatory T_reg_ populations in the spleen [72] (Table 1).

Zhang et al. showed that PLGA nanoparticles containing Rapamycin and blood clotting factor FVIII were much more effective than exogenous FVIII in Hemophilia A therapy (where lack of FVIII results in a bleeding disorder) [73] (Table 1). Exogenous FVIII administration fails in the long term because of the development of FVIII-specific antibodies from B cells. Poly(D,L-lactide-co-glycolic) acid or PLGA nanoparticles are FDA-approved organic (copolymer) nanoparticles that are used for drug delivery in vivo and are popular due to their material biocompatibility and biodegradability. For transplant tolerance induction, PLGA nanoparticles have been used to reduce IR injury by delivering insoluble drugs to hepatocytes [74] and targeting dendritic cells in grafts to improve graft survival [75]. PLGA nanoparticles containing Rapamycin tolerized B cells against FVIII on the nanoparticles [73].

In a single MHC-mismatched murine model of skin transplantation, Shahzad et al. prepared three i.v. infusions of PLGA-nanoparticles, coated with target donor alloantigen H-2K^b^-Ig dimer, modulators anti-Fas mAb, PD-L1-Fc, TGF-β (to induce apoptosis, inhibit activation and proliferation of targeted cells and induce Tr_egs_) and CD47-Fc to inhibit phagocytosis from macrophages. They showed that the nanoparticles could specifically target and deplete donor antigen-specific CD8^+^ T cells in the graft, spleen and peripheral blood (>90% reduction compared to blank nanoparticles), thereby increasing the survival of previously implanted skin allograft [76]. Larger 200 nm nanoparticles were observed to be more effective than their smaller 80 nm counterparts. It is likely that the larger surface area in the 200 nm nanoparticles resulted in a higher density of coated molecules and increased the contact area with alloantigen-specific CD8^+^ T cells. The authors verified with immunofluorescence staining that the coated nanoparticles mostly co-localized with CD8^+^ T cells compared to other immune cell populations suggesting a direct contact-based apoptotic depletion. Additionally, through measurement of recipient T cell populations, tumor challenge experiments, biochemical tests and organ histopathology, it was confirmed that the recipient’s immune system was intact and organ functions were comparable to untreated controls [76] (Table 1).

Although the skin allograft model may be a weak immunogenic transplant model, the significant increase in mean graft survival time compared to untreated controls through direct alloantigen-specific CD8^+^ T cell depletion was a novel finding and is worth further investigation for its applications to organ transplantation. Hlavaty et al. showed nanoparticle-based induction of chimerism in a minor histocompatibility antigen sex-mismatched murine model of bone marrow transplant [77]. In this model, male C57/Bl/6 donor bone marrow was rejected by the female recipient due to the donor Hy peptide antigens CD4 epitope Dby, recognized by CD4^+^ and CD8^+^ recipient T cells. However, when the Dby peptide was grafted or encapsulated in 500 nm poly(lactide-co-glycolide; PLG) nanoparticles and delivered i.v. on days 7 and 1 (day 0 is the day of the transplant) to low-dose irradiated female recipient mice, tolerance to male bone marrow was induced, as evaluated by the percentage of donor CD45.1^+^ CD90.2^+^ T cells in the peripheral blood samples of female mice. Long-term graft survival was observed to be comparable to female bone marrow donor controls. The authors also found that the tolerance induction was reduced by PD-1 blockade and that natural T_regs_ were unnecessary to induce tolerance. An important finding was that only 1.5 μg of peptide engrafted to nanoparticles was required i.v. for tolerance induction vs. 300 μg of free peptides delivered intranasally. Thus, the nanoparticles provide a 200× dose enhancement [77] (Table 1). Although the authors discussed the role of tolerogenic DCs in tolerance induction, this needs to be investigated further and studied for other allograft models, including the liver.

While immunosuppressive therapy has proven paramount to transplant success, lifelong systemic use has often led to poor patient compliance worsening morbidity, mortality and graft survival. Tolerance protocols have demonstrated immunological pathways to prevent T cell response and chronic rejection. In this manuscript, apart from pre-transplant conditioning using nanoparticles, we also consider post-transplant use of nanoparticles that specifically support the delivery of tolerance-inducing medications in the initial post-operative period.

Targeted and controlled drug delivery carriers have played a fundamental role in individualizing drug-dependent therapies. Drug targeting and controlled administration have been widely investigated, employing the novel routes offered by nanotechnology, including injection and implantation. A study conducted using doxorubicin in polymeric micelles, then in multistage nanovectors, demonstrated the toxicity to normal cells was significantly reduced [78]. Furthermore, by conjugating with receptor binding to antibody onto multistage nanovectors, particles have demonstrated to display significant adhesions to targeted binding spots. The functionalization of these nanovectors with cellular membrane proteins provides a platform for nanoparticles to avoid opsonization and macrophage uptake [79,80,81]. Such multistage nanovectors, functionalized with targeting ligands and loaded with therapeutic cargo (e.g., Rapamycin), are worth investigating for post-transplantation tolerance induction.

Similarly, targeted and controlled drug delivery using nanoparticles plays fundamental roles in mitigating immunosuppression toxicity. Nanocarriers have also proven to be a promising platform to achieve tolerogenic antigen presentation by delivering antigens of interest to specific cell types [77,82,83]. A combination of antigens and immunological agents provides an excellent tool for tolerance induction in the post-operative phase of liver transplants.

### 2.4. Preconditioning with Hyperthermia Can Avoid Chemotherapy Toxicity

Current conditioning methods to expand the donor liver pool include normothermic perfusion, in which a perfusate buffer solution containing hematocrit and pharmacological agents is circulated through the liver to condition and reverse injury in extended criteria livers, including steatotic livers [84,85,86]. Steatosis is the accumulation of lipid droplets within the hepatocytes. Macrosteatotic livers (nuclei of hepatocytes displaced due to the presence of lipid droplets), more than microsteatotic livers (nuclei of hepatocytes not displaced by lipid droplets) are subject to higher lipid peroxidation [87,88,89,90] and more exuberant IRI and proinflammatory responses with the release of tumor necrosis factor (TNF-α) [90,91] and neutrophil infiltration [92]. Pharmacological approaches to defatting the liver improve graft survival and lower injury associated with IRI [87,88,92]. However, these interventions can be time-consuming (9 h to 7 days) and costly and have systemic side effects.

#### Preconditioning with Whole-Body Hyperthermia

The hypothalamus regulates body temperature, and fever is its response to disease, infection or injury-induced inflammation. Although the increase in core body temperature >38 °C is broadly termed “hyperthermia,” this is distinguished from applying energy to raise tissue temperature (41–45 °C) when treating diseases such as cancer. At a molecular level, whole-body hyperthermia upregulates HSPs and other protective molecules. HSP70 expression and extracellular release also induce the expression of inflammatory cytokines such as IL-6 and TNF-α, which are required for liver regeneration. Short whole-body hyperthermic treatments (42–42.5 °C for 10 min) of donor rats with fatty livers prior to liver transplantation showed improved survival in the recipient rats transplanted with livers from the hyperthermia-conditioned cohorts (80% vs. 10% in control). This correlated with increased expression of HSP70 in the liver grafts in a time-dependent manner. Furthermore, the release of TNF-α and IL-10 was also suppressed in these recipients [93]. Rats treated with whole-body hyperthermia also exhibited better tolerance to damage from warm IRI [94]. Thus, whole-body hyperthermic conditioning has shown potential for a translational alternative for preconditioning.

In experiments comparing responses in wild-type (WT) mice vs. HSP70 knockout mice in a partial hepatectomy following whole-body hyperthermia, WT mice displayed upregulated HSP70, IL-6 and TNF-α with competent liver regeneration. On the other hand, liver regeneration was impaired in HSP70 knockout mice [95]. Similarly, in human left lobe living donors (LDLT), it was observed that HSP70 was upregulated during early liver regeneration [95]. Experiments probing whether HSP70 upregulation stimulated by whole-body hyperthermia is sufficient for liver regeneration would be valuable.

Heat-induced HSP upregulation in cells depends on the time of exposure at elevated temperatures defined by the Arrhenius relationship [96]. In addition, whole-body hyperthermic preconditioning in preclinical models (normal and steatotic livers) reduces IRI following recovery by upregulating HSP expression [93,97]. However, a rise in core body temperature >42 °C can be fatal. Additionally, specific physiological, metabolic and molecular effects observed in livers during whole-body hyperthermia could have deleterious effects as they are mediated by the central nervous system instead of being regulated locally [98]. Thus, local graft treatment with hyperthermia is an attractive route and should be investigated for its efficacy in improving functional and transplantation outcomes.

### 2.5. Whole-Body vs. Local Hyperthermia: the Case for Expanding the Liver Donor Pool by Conditioning with Nanoparticle Hyperthermia

Mild local (liver-specific) hyperthermia in pigs and humans increases metabolic activity through increased glycogenolysis, increased lipolysis and temperature-dependent mitochondrial respiration while maintaining viability [98,99]. Although mechanistic details are unclear, evidence suggests that HSP upregulation by local hyperthermic perfusion [100,101] and (magnetic) nanoparticle hyperthermia [51,52] would confer protection to grafts and facilitate studies to elucidate mechanisms because they offer more precisely targeted heat deposition in liver grafts with spatiotemporal control and image-guided validation. Magnetic nanoparticles suspended in fluid generate heat when exposed to an alternating magnetic field, raising the temperature of the fluid as it is perfused through the tissue. Localized thermal treatments may further enable spatiotemporal control of HSP expression while simultaneously minimizing systemic side effects associated with whole-body hyperthermia.

Short (10–15 min) ex vivo hyperthermic machine perfusion of liver grafts isolated from donor rats resulted in better functional outcomes, including bile production, less mitochondrial enzyme loss and upregulated HSP expression [102]. Ex vivo machine perfusion can offer greater control of HSP expression through localized delivery of heated perfusate to the graft and local control of temperature, to which expression of HSPs is intimately tied.

Diller et al. showed that hyperthermia-induced HSPs (HSP27, HSP60 and HSP70) in prostate cancer cells could be modeled using a non-linear differential equation and appropriate experimental fitting parameters to predict thermally induced HSP expression [103]. Thorne et al. [98] discuss the potential benefits of hyperthermia-induced vasodilation in the liver, which could reduce ROS-based damage through increased blood perfusion.

Combining the protective effects of hyperthermia with the selectivity of ex vivo machine perfusion and temperature control offered by magnetic hyperthermia technology, one can explore a targeted approach to precondition livers prior to transplant. Temperature- and pressure-controlled ex vivo perfusion can be used to control the perfusate composition and temperature. Targeted perfusion systems are already in clinical use (hyperthermic intraperitoneal chemotherapy, or HIPEC) for treating peritoneal cancers [104,105], and such technologies can be suitably adapted to condition livers.

Additionally, temperature control on the perfusate can be exercised non-invasively by innovative use of magnetic hyperthermia technology by adding magnetic nanoparticles to the perfusate and the application of alternating magnetic fields to the liver placed in a radiofrequency coil (RF; Figure 1) [51,106]. RF refers to the frequency of applied alternating magnetic field (AMF) to generate hysteresis-based heating from magnetic nanoparticles. Although the RF encompasses a broad frequency range ranging from 20 kHz to 300 GHz, for MHT applications, this is typically in the 100 kHz–1 MHz frequency range. This would enable the improved modeling of HSP expression kinetics and the validation of nanoparticle localization through imaging modalities such as MRI and MPI. The washout of nanoparticles from livers has been demonstrated in rat models during hypothermic perfusion [55]. The experimental validation of washout during hyperthermic treatments is needed, as the hypermetabolic state of cells during hyperthermia could result in some endocytosis of nanoparticles, depending on the treatment time and flow velocity. The correlation of HSP expression with treatment schedules and tolerance induction also needs experimental validation. Additional research on the effects of persistent overexpression of HSPs and genetic expression of HSPs (vs. those which are thermally induced) is required to optimize the reduction of IRI. Targeting intercellular adhesion molecule-1 (ICAM-1) [92] binding with nanoparticles [49], largely unexplored, could prevent the binding of neutrophils and potentially lower IRI. Through combined ICAM-1 blockade and hyperthermic localized ex vivo perfusion with nanoparticles, greater control over the hypermetabolic state and lowering of the lipid content in hepatocytes in fatty livers could be possible in a shorter time, thus increasing the pool of transplantable livers.

Magnetic nanoparticles with a well-characterized specific loss power vs. an applied magnetic field amplitude allow greater control over the thermal dose delivered to the graft and can help elucidate quantitative relationships between temperature, exposure time and graft tolerance. Although the use of MHT for tolerance induction is not well studied, we think that there is an opportunity to leverage the control over thermal energy that MHT allows for investigating protocols for HSP upregulation and tolerance induction.

**Table 1 ijms-24-07496-t001:** Summary of representative literature for nanoparticle-based tolerance induction in various transplant models.

Study	Tissue/Model	Nanoparticle	Functionality	Results
Tietjen et al. [48]	Human kidneys.	PLA-PEG nanoparticles, 170 nm mean diameter.	Anti-CD31 conjugated nanoparticles to target endothelial cells.	5- to 10-fold enhancement of localization of nanoparticles vs. unconjugated nanoparticles.
Cui et al. [50]	Human umbilical vein endothelial cells (HUVECs), arterial allografts.	Poly (amine co-ester) nanoparticles, 288 nm mean diameter.	Loaded with non-self MHC II specific siRNA.	Attenuation of MHC II molecules, reduced T cell infiltration and T cell-mediated inflammation and improved allograft histology.
Zhu et al. [56]	Mouse aortic and tracheal allografts.	Polyethylene glycol micelles, 15.3 nm mean diameter.	Rapamycin (tolerogenic drug) loaded	Reduced the secretion of inflammatory cytokines and prevented allograft rejection post-transplantation with a 10-fold lower rapamycin dose vs. free Rapamycin.
Stead et al. [72]	Murine and non-human primate (marmosets) dendritic cells in vivo targeting.	Porous silicon nanoparticles, 21 nm mean diameter.	Nanoparticles coated with DC-specific intercellular adhesion molecule-3 grabbing non-integrin (DC-SIGN), monoclonal antibody CD11c, ovalbumin (OVA) and loaded with Rapamycin.	Upregulated donor-specific regulatory Treg populations in the spleen.
Zhang et al. [73]	Hemophilia A C57BL/6 mice in vivo therapy.	PLGA nanoparticles.	Nanoparticles containing rapamycin and blood clotting factor FVIII	Tolerized B cells against FVIII on the nanoparticles, thereby more effective vs. free FVIII.
Shahzad et al. [76]	Single MHC-mismatched murine model of skin transplantation	PLGA nanoparticles, 80 and 200 nm.	Nanoparticles coated with target donor alloantigen H-2Kb-Ig dimer, modulators anti-Fas mAb, PD-L1-Fc, TGF-β (to induce apoptosis, inhibit activation and proliferation of targeted cells and induce Tregs) and CD47-Fc to inhibit phagocytosis from macrophages.	Nanoparticles could specifically target and deplete donor antigen-specific CD8^+^ T cells in the graft, spleen and peripheral blood (>90% reduction compared to blank nanoparticles), thereby increasing the survival of previously implanted skin allograft.
Hlavaty et al. [77]	Sex-mismatched murine model of bone marrow transplant.	Poly(lactide-co-glycolide; PLG) nanoparticle, 500 nm mean diameter.	Donor Hy peptide antigens CD4 epitope Dby, grafter nanoparticles.	Coated nanoparticles provide a 200-fold dose enhancement vs. free peptide in inducing tolerance to male bone marrow.

## 3. Advantages and Disadvantages of Nanotechnology

Nanotechnology is determined to be advantageous over conventional and current chemotherapeutic interventions for the following reasons. Nanoparticles allow a non-invasive route to condition donor grafts and induce transplantation tolerance in recipients.

They allow high drug encapsulation efficiency and drug stability, especially for insoluble drugs that cannot be delivered in free form. For example, Rapamycin has demonstrated effective immunosuppression; however, its water insolubility makes it challenging to develop formulations [107]. Several publications cited in this review demonstrate the enhanced stability and efficacy of Rapamycin in a nanoparticle conjugate [55,71,72].

The optimal choice of nanoparticle material (organic and inorganic) and biocompatible coatings (e.g., PEG, starch, lipid) and ligands on the nanoparticle can improve spatial localization, reduce uptake by the mononuclear phagocyte system (MPS), protect the drug from premature degradation and control temporal release of the drugs to the graft, thereby lowering systemic side effects. Nanochannel membranes in mesoporous nanoparticles can offer a constant, sustained release of immunosuppression drugs and can be tuned in channel sizes of 2–200 nm [108] for receptor blockade. They also offer in vivo delivery system periods ranging from 1–6 months [109]. The “targeting” advantage is more pronounced when combined with ex vivo perfusion methods in organs versus in vivo delivery, where ex vivo delivers the NPs directly to the graft.

Gold, iron oxide and quantum dots are often used as contrast agents in nanoparticle drug formulations to assist in delineating anatomy and physiology for imaging, enabling validation of drug delivery localization. The use of nanoparticles has exhibited a six-fold contrast enhancement over conventional agents [110].

Magnetic nanoparticles which are responsive to external magnetic fields can be used to improve the spatial localization of drug-nanoparticle conjugates to the graft through the use of magnetic field gradients [111]. The use of magnetic nanoparticles in liposomal drug nanoparticle formulations allows the temperature-controlled release of drugs through the application of alternating magnetic fields (AMF), which heat the magnetic nanoparticles through hysteresis heating. When the nanoparticle distribution and AMF regions are controlled, it can allow greater spatiotemporal control over the release of the drugs and their pharmacokinetics. Additionally, combination therapies involving heat and chemotherapeutics can be probed for improved efficacy in tolerance induction through potentially synergistic interactions. This multi-functionality of nanoparticles, including drug delivery, targeting, imaging and heat generation, allows the development of a truly theranostic technology for tolerance induction. Integration with NMP allows enhanced targeting of conditioning therapeutics and optimal temporal release profiles in the endothelium of the donor organ.

Inorganic nanoparticle-based drug/adjuvant delivery can increase the potency and immunogenicity of the drug, like vaccines. Nanoparticle adjuvants carrying donor antigens have been used to condition recipients [82].

Nanoparticles have demonstrated the potential to disrupt signaling pathways in T cell activation and donor antibody functions through receptor targeting that can eventually be used in place of immunosuppressive drugs [112]. Nanobodies that are therapeutic fragments of antibodies present advantages in size, stability and low immunogenic potential and can be used to stimulate inhibitory pathways and shut off immune cells to prevent allograft rejection [113,114].

While nanotechnology has many advantages, the use of nanoparticles as carriers for drugs and targeting moieties should be assessed for possible toxic side effects from long-term exposure, higher doses or retention in the organ. Thus, nanoparticles undergo extensive screening for toxicity/biocompatibility before clinical approval. Further, nanoparticles typically display altered biodistribution and pharmacokinetic properties than those of the drugs comprising their payload(s), making it imperative that the full pharmacologic profile of the nanoformulation is fully characterized.

The material composition of the nanoparticle is one such characteristic that must be carefully screened to avoid toxicity from the degradation of the materials or altered distribution in vivo. The liver, being the primary organ for detoxification in the body, isolates and eliminates various exogenic compounds through phagocytosis. Studies have demonstrated that metallic nanoparticles deposited in the liver can be responsible for hepatoxicity [115,116]. Such inorganic nanoparticles increase the concentrations of pro-inflammatory cytokines (IL-1β and IL-6) and decrease levels of anti-inflammatory cytokines (IL-10 and IL-4) [117]. The activation of pro-inflammatory cytokines can be detrimental to inducing tolerance and preventing rejection. Alongside activation of the inflammatory cytokines, inorganic nanoparticles can also induce structural changes decreasing total bilirubin and increasing alkaline phosphatase (ALP) and aspartate aminotransferase (AST), suggesting liver injury [118,119], which might lead to downstream metabolic dysfunction.

Silver nanoparticles can cause severe hepatocyte necrosis and hemorrhage, as well as multifocal peribiliary microhemorrhage and occasional portal vein endothelial damage [120]. They are generally avoided for in vivo medical uses. Many inorganic nanoparticles have also demonstrated the potential to induce hepatic inflammatory cell infiltration, increasing the density of liver collagen and initiating hepatic fibrosis with documented thickening of the Glisson capsule [121].

Gold nanoparticles have also demonstrated the potential to activate hepatic macrophages with an aggravated course of hepatitis and liver injury [122]. The mitochondrial dysfunction induced by some inorganic nanoparticles includes morphological changes with increased production of reactive oxygen species, changes in calcium content, lowered mitochondrial membrane potential, and the inhibition of various enzymes activities, electron transport chains and cellular respiration and a decline in ATP synthesis, etc. which could further lead to insufficient energy supply and affect cell viability such as apoptosis and necrosis [123,124]. Endoplasmic reticulum changes caused by inorganic nanoparticles include swelling, stress, misfolding of proteins and increasing or decreasing protein synthesis [125,126]. Clinically approved nanoparticles include organic/polymeric NPs for drug delivery, iron oxide-based nanoparticles as MRI contrast agents or a formulation combining the above modalities.

Other disadvantages can include colloidal instability of nanoparticles causing aggregation-related issues (retention in organs, occlusion in blood vessels), limited shelf life and shape and surface coating-dependent toxicity. Longitudinal studies evaluating long-term safety are, therefore, essential.

## 4. Summary and Future Outlook

Transplantation is currently the only definitive treatment for total liver failure. Gaps between supply—specifically organ availability—and unmet needs require that extended donor grafts be used. With current technology, however, the use of extended donor grafts increases the risks of graft failure because attendant IRI increases rates of rejection. There is a global unmet need to develop technologies to reduce IRI. One approach that may address this need is a technology that selectively targets receptors in the liver with nanotechnology, specifically magnetic hyperthermia. This offers the potential to block the activation of deleterious immune cascades, improving host acceptance. The additional potential for nanotechnology applications includes exploiting the immune-modulating potential of nanoparticles, which can either stimulate or suppress inflammatory responses, depending on material properties. This potentially offers substantial benefits with an appropriate rational design that must be based on data obtained from appropriate models. These opportunities will require substantially more investigation.

## Figures and Tables

**Figure 1 ijms-24-07496-f001:**
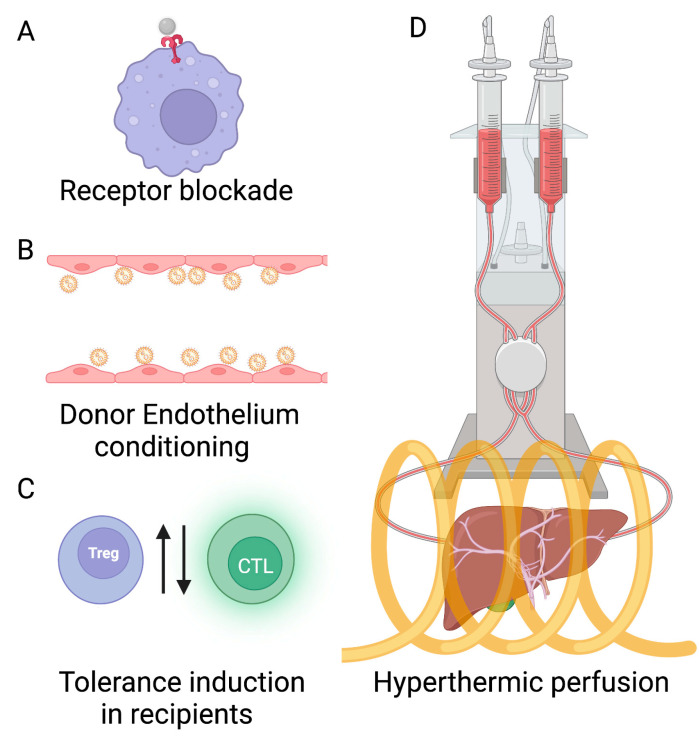
Nanotechnology offers the potential to address donor liver shortages with new approaches to organ and donor preconditioning. Immune modulation of the liver with nanoparticles can expand the donor pool through conditioning. (**A**) Receptor blockade with nanoparticles offers the potential to reduce organ-rejecting immune responses by suppressing liver-specific pro-inflammatory immune receptors. (**B**) Donor endothelium conditioning can be achieved by blocking recognition by the recipient immune system of non-self MHC molecules present on endothelial cells in the vasculature of the donor graft using nanoparticle targeting. (**C**) Tolerance induction in recipients with nanoparticles potentially improves targeting efficiency with lower doses of nanoparticle-coated donor antigens and peptides to tolerize recipient immune cells or induce chimerism. (**D**) Hyperthermic perfusion with magnetic nanoparticle hyperthermia can be used to stimulate localized HSP upregulation and controlled organ-specific preconditioning. The yellow coil is a representation of the RF coil, which will be used to apply an alternating magnetic field (AMF).

## Data Availability

No new data were created or analyzed in this study. Data sharing is not applicable to this article.

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
