# Peer review of "Nanoparticle-Based Interventions for Liver Transplantation"

_ijms, 2023, doi:10.3390/ijms24087496_

Round 1

Reviewer 1 Report

Liver transplantation is often performed in cases of acute liver failure or end-stage liver disease. The major risk for liver transplantation is ischemia/reperfusion injury. Current studies aimed to reduce cell and liver damage during ischemia/reperfusion.

In the current manuscript, the authors describe nanotechnology as a potential strategy to improve the success of liver transplantation. However, the authors should better emphasize this potential, by describing the advantages over current interventions. In addition, the final sections need to be greatly improved as they can be confusing and speculative.

Major points:

1. What are the advantages of nanotechnology over current interventions?

2. Are nanotechnologies only advantageous during perfusion? Or can they be used in different stages of liver transplantation? For example, post-transplant?

3. Are there disadvantages to nanotechnology?

4. Section 2.5 is confusing and speculative. There is very little literature regarding magnetic hyperthermia in the context of liver transplantation. Also, the use of nanotechnology alongside hyperthermia is not made clear.

Minor points:

1. There are some references missing, for example:

- “HMGB1 is released from injured hepatocytes to stimulate liver non-parenchymal 146 cells such as Kupffer cells via TLR-4 signaling.” (line 146)

- “An additional benefit is that magnetic hyperthermia can enhance tolerance in both graft tissue and recipient by targeted immunomodulation” (line 421)

2. A table would be helpful to summarize sections 2.1, 2.2 and 2.3.

Author Response

We thank the reviewers for taking time to provide their thoughtful and insightful comments improve the manuscript. We have provided responses to the reviewer comments below in italicized text and have made changes to the manuscript text in response to comments.

Reviewer 1

Liver transplantation is often performed in cases of acute liver failure or end-stage liver disease. The major risk for liver transplantation is ischemia/reperfusion injury. Current studies aimed to reduce cell and liver damage during ischemia/reperfusion.

In the current manuscript, the authors describe nanotechnology as a potential strategy to improve the success of liver transplantation. However, the authors should better emphasize this potential, by describing the advantages over current interventions. In addition, the final sections need to be greatly improved as they can be confusing and speculative.

Major points:

  1. What are the advantages of nanotechnology over current interventions?

Nanotechnology is advantageous over conventional chemotherapeutic and surgical interventions for the following reasons -  

  • Nanoparticles allow a non-invasive means to condition donor grafts and induce transplantation tolerance in recipients.
  • Nanoparticles allow high drug encapsulation efficiency and drug stability, especially for insoluble drugs that cannot be delivered in free form. Eg. Rapamycin has demonstrated effective immunosuppression. However, its water insolubility has made it challenging to develop formulations [108]. Several publications cited in this review demonstrate the enhanced stability and efficacy of rapamycin in a nanoparticle conjugate [56,72,73].
  • Optimal choice of nanoparticle material (organic, inorganic) and biocompatible coatings (eg., PEG, starch, lipid) and ligands on the nanoparticle can improve spatial localization, reduce uptake by the mononuclear phagocyte system (MPS), protect the drug from premature degradation and control temporal release of the drugs to the graft, thereby lowering systemic side effects. Eg. Nanochannel membranes in mesoporous nanoparticles can offer a constant, sustained release of immunosuppression drugs and can be tuned in channel sizes of 2-200nm [109] for receptor blockade. They also offer in vivo deliver system periods ranging from 1- 6 months [110]. The “targeting” advantage is obviously more pronounced when combined with ex vivo perfusion methods in organs vs in vivo delivery, where ex vivo delivers the NPs directly to the graft [ref].
  • Magnetic nanoparticles responsive to external magnetic fields can be used to improve spatial localization of the drug-nanoparticle conjugates to the graft through the use of magnetic field gradients [111].
  • Gold, iron oxide and quantum dots are often used as contrast agents in nanoparticle drug formulations to assist delineating anatomy and physiology for imaging, enabling validation of drug delivery localization. The use of nanoparticles has exhibited a 6-fold contrast enhancement over conventional agents [112].
  • Use of magnetic nanoparticles in liposomal drug nanoparticle formulations allows temperature-controlled release of drugs through application of alternating magnetic fields (AMF) which heat the magnetic nanoparticles through hysteresis heating. When the temperature stimulus and AMF regions is controlled, it can allow greater spatiotemporal control over release of the drugs and their pharmacokinetics. Additionally, combination therapies involving heat and chemotherapeutics can be probed for improved efficacy in tolerance induction through potentially synergistic interactions.
  • This multifunctionality of nanoparticles including drug delivery, targeting, imaging and heat generation allows the development of a truly theranostic technology for tolerance induction.
  • Inorganic nanoparticles-based drug/adjuvant delivery can increase the potency and immunogenicity of the drug, like vaccines. Nanoparticle adjuvants carrying donor antigens have been used to condition recipients [83].
  • Nanoparticles have demonstrated ability to disrupt signaling pathways in T cell activation and donor antibody functions through receptor targeting that can eventually be used in place of immunosuppressive drugs [113].
  • Nanobodies that are therapeutic fragments of antibodies present advantage in size, stability and low immunogenic potential and can be used to stimulate inhibitory pathways and shut off immune cells to prevent allograft rejection [114,115].
  • Integration with MNP allows enhanced targeting of conditioning therapeutics and optimal temporal release profiles in the endothelium of the donor organ.
  1. Are nanotechnologies only advantageous during perfusion? Or can they be used in different stages of liver transplantation? For example, post-transplant?

      We thank the reviewer for their thoughtful comment on the use of nanoparticles not only prior to transplant but also post-transplant. While immunosuppressive therapy has proven paramount to transplant success, lifelong systemic use has often led to poor patient compliance worsening morbidity, mortality, and graft survival. Tolerance protocols have demonstrated immunological pathways to prevent T cell response and chronic rejection. In this manuscript apart from pre-transplant conditioning using nanoparticles, we also target post-transplant use of nanoparticles that specifically support the delivery of tolerance inducing medications in the initial post-operative period.

Targeted and controlled drug delivery carriers have played a fundamental role in individualizing drug dependent therapies. Drug targeting and controlled administration have been widely investigated, employing the novel routes offered by nanotechnology, including injection and implantation. A study conducted using doxorubicin into polymeric micelles, then into multistage nanovectors demonstrated the toxicity to normal cells was significantly reduced [78]. Furthermore, by conjugating with receptor binding to antibody onto multistage nanovectors, particles have demonstrated ability to display significant adhesions to targeted binding spots. The functionalization of these nanovectors with cellular membrane proteins provides a platform for nanoparticles to avoid opsonization and macrophage uptake [79-81]. Such multistage nanovectors functionalized with targeting ligands and loaded with therapeutic cargo (eg. rapamycin) are worth investigating for post-transplantation tolerance induction.

Similarly, targeted and controlled drug delivery using nanoparticles play fundamental roles in mitigating immunosuppression toxicity. Nanocarriers have also proven to be promising platform to achieve tolerogenic antigen presentation by delivering antigens of interest to specific cell types [77,82,83]. A combination of antigens and immunological agents provide an excellent tool for tolerance induction in the post-operative phase of liver transplant.

  1. Are there disadvantages to nanotechnology?

While nanotechnology has many advantages, the use of nanoparticles as carriers for drugs and targeting moieties should be assessed for possible toxic side effects from long term exposure, higher doses or retention in the organ. Thus, nanoparticles undergo extensive screening for toxicity/biocompatibility before clinical approval.

Material composition of the nanoparticle is one such characteristic that must be carefully screened for, to avoid toxicity from degradation of the materials in vivo. The liver being the primary organ for detoxification in the body, isolates and eliminates various exogenic compounds through phagocytosis. Studies have demonstrated that the metallic nanoparticles deposited in the liver can results in hepatoxicity [116,117]. Such inorganic nanoparticles increase the concentrations of pro-inflammatory cytokines (IL-1β and IL-6) and decrease the levels of anti-inflammatory cytokines (IL-10 and IL-4) [118]. Activation of pro-inflammatory cytokines would be detrimental to inducing tolerance and prevent rejection. Alongside activation of the inflammatory cytokines, inorganic nanoparticles can also cause structural changes decreasing total bilirubin and increasing alkaline phosphatase (ALP) and aspartate aminotransferase (AST) suggesting liver injury [119,120] which might lead to downstream metabolic dysfunction. Silver nanoparticles have demonstrated to cause severe hepatocyte necrosis and hemorrhage as well as multifocal peribiliary microhemorrhage and occasional portal vein endothelial damage [121]. Many inorganic nanoparticles have also demonstrated to induce hepatic inflammatory cell infiltration, increasing the density of liver collagen and initiating hepatic fibrosis with documented thickening of the Glisson capsule [122]. Gold nanoparticles have also demonstrated to activation of hepatic macrophages with an aggravated course of hepatitis and liver injury [123]. Mitochondrial dysfunction enticed by some inorganic nanoparticles include morphological changes with increased production of reactive oxygen species, changes in calcium content, descending mitochondrial membrane potential and inhibition of various enzymes activities, inhibiting electron transport chains, inhibition of cellular respiration, decline in ATP synthesis, etc. which could further lead to insufficient energy supply and affect cell viability such as apoptosis and necrosis [124,125]. Endoplasmic reticulum changes caused by inorganic nanoparticles include swelling, stress, misfolding of proteins and increasing or decreasing protein synthesis [126,127]. Clinically approved nanoparticles include organic/polymeric NPs for drug delivery, iron oxide-based nanoparticles as MRI contrast agents or a formulation combining the above modalities.

Other disadvantages can include colloidal instability of nanoparticles causing aggregation related issues (retention in organs, occlusion in blood vessels), limited shelf life and shape and surface coating dependent toxicity. Longitudinal studies evaluating long-term safety is, therefore, essential.

  1. Section 2.5 is confusing and speculative. There is very little literature regarding magnetic hyperthermia in the context of liver transplantation. Also, the use of nanotechnology alongside hyperthermia is not made clear.

We have reworded this paragraph to emphasize the future potential of magnetic hyperthermia in inducing transplantation tolerance based on the current literature showing improved transplantation outcomes following heat shock to the graft. Magnetic nanoparticles with well characterized specific loss power vs. applied magnetic field amplitude allow greater control over the thermal dose delivered to the graft and can help elucidate quantitative relationships between temperature, exposure time and graft tolerance. Although the use of MHT for tolerance induction is not well studied, we think that there is an opportunity to leverage the control over thermal energy that MHT allows for investigating protocols for HSP upregulation and tolerance induction.

Minor points:

  1. There are some references missing, for example:

- “HMGB1 is released from injured hepatocytes to stimulate liver non-parenchymal 146 cells such as Kupffer cells via TLR-4 signaling.” (line 146)

Reference was added [42].

- “An additional benefit is that magnetic hyperthermia can enhance tolerance in both graft tissue and recipient by targeted immunomodulation” (line 421).

Reference was added. Sentencing was rephrased.

  1. A table would be helpful to summarize sections 2.1, 2.2 and 2.3.

We have now added a table that summarizes sections 2.1-2.3.

Reviewer 2 Report

This paper reviews the potential benefit of nanotechnology to extend the donor criteria and to fight the shortage of grafts.

It covers an interesting topic and it is well written. 

Some paragraphs (especially 2.1 and  2.3) are quite long and might be difficult to catch up. I would suggest to add more figures and tables to summarize and to show the content of the text. 

Author Response

We thank the reviewers for taking time to provide their thoughtful and insightful comments improve the manuscript. We have provided responses to the reviewer comments below in italicized text and have made changes to the manuscript text in response to comments.

Reviewer 2

This paper reviews the potential benefit of nanotechnology to extend the donor criteria and to fight the shortage of grafts.

It covers an interesting topic and it is well written. 

Some paragraphs (especially 2.1 and  2.3) are quite long and might be difficult to catch up. I would suggest to add more figures and tables to summarize and to show the content of the text. 

We have now added a table that summarizes sections 2.1-2.3.

Reviewer 3 Report

This is a review on the possible role of nanoparticles intervention in the liver transplantation.

I have some comments.

1.         (Figure 1) It should be described that the orange coil in the figure indicated a RF coil in the legend.

2.         (L275) PLGA needs an explanation.

3.         (L398) RF needs an explanation.

Author Response

We thank the reviewers for taking time to provide their thoughtful and insightful comments improve the manuscript. We have provided responses to the reviewer comments below in italicized text and have made changes to the manuscript text in response to comments.

Reviewer 3

This is a review on the possible role of nanoparticles intervention in the liver transplantation.

I have some comments.

  1. (Figure 1) It should be described that the orange coil in the figure indicated a RF coil in the legend.

We have added description to the figure and caption.

  1. (L275) PLGA needs an explanation.

Poly(D,L-lactide-co-glycolic) acid or PLGA nanoparticles are FDA-approved organic (copolymer) nanoparticles that are used for drug delivery in vivo and are popular due to their of material  biocompatibility and biodegradability. For transplant tolerance induction, PLGA nanoparticles have been used to reduce IR injury by delivering insoluble drugs to hepatocytes [75], and target dendritic cells in grafts to improve graft survival [76].

  1. (L398) RF needs an explanation.

Radiofrequency, or RF, refers to the frequency of applied alternating magnetic field (AMF) to generate hysteresis-based heating from magnetic nanoparticles. Although the RF encompasses a broad frequency range ranging from 20 kHz to 300 GHz, for MHT applications this is typically in the 100 kHz to 1 MHz frequency range.

Round 2

Reviewer 1 Report

All my comments/suggestions have been taken into account.